# *Ziziphus mauritiana* Lam. Bark and Leaves: Extraction, Phytochemical Composition, *In Vitro* Bioassays and *In Silico* Studies

**DOI:** 10.3390/plants13162195

**Published:** 2024-08-08

**Authors:** Kouadio Ibrahime Sinan, Stefano Dall’Acqua, Stefania Sut, Abdullahi Ibrahim Uba, Ouattara Katinan Etienne, Claudio Ferrante, Jamil Ahmad, Gokhan Zengin

**Affiliations:** 1Physiology and Biochemistry Laboratory, Department of Biology, Science Faculty, Selcuk University, Konya 42130, Turkey; nelofarkhattak@gmail.com (N.); sinankouadio@gmail.com (K.I.S.); gokhanzengin@selcuk.edu.tr (G.Z.); 2Department of Pharmacy, Botanic Garden “Giardino dei Semplici”, Università degli Studi “Gabriele d’Annunzio”, 66100 Chieti, Italy; claudio.ferrante@unich.it; 3Department of Pharmaceutical and Pharmacological Sciences, University of Padova, 35131 Padua, Italy; 4Department of Molecular Biology and Genetics, Istanbul AREL University, Istanbul 34537, Turkey; 5Laboratoire de Botanique et Valorisation de la Diversite Vegetale, UFR Science de la Nature, Universite Nangui Abrogoua, 22 BP 1428 Abidjan 22, Abidjan 83111, Côte d’Ivoire; katinan.etienne@gmail.com; 6Department of Human Nutrition, The University of Agriculture Peshawar, Peshawar 25000, Khyber Pakhtunkhwa, Pakistan; jamil.hn@aup.edu.pk

**Keywords:** antioxidant, enzyme inhibition, homogenizer-assisted extraction, in silico analysis, multivariate analysis, phenolics, solvents

## Abstract

In this work, homogenizer-assisted extraction (HAE) and maceration (MAE) were applied on leaves and bark of *Ziziphus mauritiana* using water and methanol (MeOH) as solvents. HAE and MAE extracts were compared through liquid chromatography coupled with mass spectrometry (LC-MS) and evaluating the antioxidant activity, and enzyme inhibition against acetylcholinesterase (AChE), butrylcholinesterase (BChE), tyrosinase, α-amylase, and α-glucosidase. Considering the phytochemical contents and the bioassays results, the HAE extracts resulted favorably with larger content of phenolics and higher antioxidant activity. The MeOH extracts displayed the highest α-amylase inhibitory activity, with HAE MeOH leaf extract leading at 0.78 mmol acarbose equivalent (ACAE)/g. In conclusion, the study highlights that HAE can increase the extraction of phenolic and flavonoid from *Z. mauritiana* plant materials compared to maceration. Further research could explore the potential therapeutic applications of *Z. mauritiana* extracts, especially HAE MeOH leaf extracts, for their notable antioxidant and enzyme inhibitory activities, facilitating the way for the development of novel pharmaceutical interventions.

## 1. Introduction

Free radicals are consistently generated as natural byproducts of oxygen metabolism during mitochondrial oxidative phosphorylation, the mitochondrion one being the primary origin of free radicals [1,2]. The involvement of free radicals in numerous biochemical reactions as well as in disease states has been thoroughly established [3,4]. Unbalanced oxidative stress is related to various disease conditions, such as cerebrovascular disease, cancer, arteriosclerosis, heart disease, ulcers, osteoporosis, rheumatoid arthritis, diabetes mellitus, neurodegenerative diseases (e.g., Parkinsonism and Alzheimer’s disease), and many more [4,5]. In recent years, significant attention has been directed toward exploring the potential therapeutic benefits of antioxidants in managing degenerative diseases associated with substantial oxidative damage. Numerous plant extracts and various classes of phytochemicals have demonstrated noteworthy antioxidant properties [6,7,8], and plants of the genus *Ziziphus* (Family: Rhamnaceae) are well known in this regard. *Ziziphus* species exhibit growth as either shrublets, shrubs, or trees characterized by thorny branches [9]. The genus *Ziziphus* includes about 100 deciduous or evergreen tree and shrub species found in tropical and subtropical regions [9]. Certain species, such as *Z. mauritiana* Lam. and *Z*. *spinachristi* (L.) wild, are found on nearly every continent. These species are known for their highly nutritious fruits that are typically consumed fresh [10]. Various plant parts of *Ziziphus* have a history of traditional use in treating conditions such as asthma, allergies, depression, bronchial disorders, measles, and ulcers [11,12,13,14,15].

*Z*. *mauritiana* Lam. is a spiny shrub native to southern Asia and eastern Africa [10], and distribution greatly expanded by humans, also introduced in America. The plants are shrubs or trees up to 15 m tall, spiny or rarely unarmed, and have gray and fissured barks. It has a blade ovate to elliptic or suborbicular leaves [16]. The leaves are applied to treat sores, and the roots are employed to prevent and cure skin diseases. This plant is a rich source of proteins, fats, dietary fiber, and various inorganic elements such as calcium, phosphorus, magnesium, potassium, sodium, chlorine, and sulfur [9,17]. Additionally, the aerial parts of the plant contain cetyl alcohol, alkaloids like protopine and berberine, flavonoids such as quercetin and kaempferol, and sterols including sitosterol, stigmasterol, lanosterol, and diosgenin [9,18].Various studies have demonstrated a high antioxidant capacity in *Z*. *jujuba* under different physiological conditions [19,20].

Extraction is a crucial step in the production of plant extracts. In this sense, in recent years, it has been reported that several green extraction techniques, such as microwave- or ultrasound-assisted extractions, are being replaced by traditional techniques such as maceration and Soxhlet [21,22]. Among green extraction techniques, homogenizer-assisted extraction (HAE) is gaining increasing interest in phytochemical studies. HAE is a method in which the sample is rapidly rotated, introduced into the dispersing head in a straight line, and then pushed outward through slots in the rotor assembly. HAE has several advantages. For example, in this process, the amount of solvent used is lower than in other techniques and thus ensures lower energy consumption and a shorter extraction time. This also provides reducing the particle size of the plant material, which facilitates the release of phytochemicals into the medium [23].

The antioxidant capacity of a plant extract can be related to the plant part used, the extraction method, and the type of solvent used for extraction [24]. Different extraction methods and solvents may yield results with low correlation due to variations in antioxidant mechanisms. Moreover, the choice of antioxidant capacity analysis depends on the free radical generator or oxidant, as well as the applied technology [25]. Therefore, comparing different antioxidant methods can provide valuable insights into understanding the relationship between the bioactive compound profiles in different parts of the plant and their antioxidant activity.

In this work, we aimed to determine the differences between *Z. mauritiana* leaf and bark extracts using homogenization and maceration techniques in terms of chemical composition and biological activities (antioxidant and enzyme inhibitory properties). The results were also evaluated using multivariate and in silico analyses to gain further insights. The obtained results could shed light on the biopharmaceutical potential of *Z. mauritiana.*

## 2. Results and Discussion

### 2.1. Total Phenolics and Flavonoids Content

The results of spectrophotometrical assays are reported in Table 1 and show that *Z*. *mauritiana* is rich in phenolics and flavonoids. In the comparison of the two extraction methods, homogenizer-assisted extraction (HAE) using MeOH as the solvent yields higher total phenolic content (TPC) than macerated-assisted extraction (MAC) with methanol (MeOH). Specifically, for leaves, HAE-MeOH extracts contain 112.01 mg GAE/g of phenolics, while MAC MeOH extracts contain 100.24 mg GAE/g. For bark, HAE MeOH extracts contain 105.99 mg GAE/g, while MAC MeOH extracts contain 101.02 mg GAE/g. Furthermore, the choice of solvent also impacts the TPC. In MAC extracts, MeOH results in higher phenolic content compared to water extracts. However, in HAE extracts, the water extract yields a higher TPC than MeOH. For example, in HAE, water extracts from leaves exhibited 49.26 mg GAE/g, whereas MAC water extracts contain 34.57 mg GAE/g. Similarly, for bark, HAE water extracts contain 45.48 mg GAE/g, while MAC water extracts contain 42.78 mg GAE/g of phenolics.

A similar pattern is observed in the TFC as well. Among the leaves extracts, HAC MeOH extracts showed the highest TFC at 51.91 mg RE/g, followed by MAC MeOH extracts at 48.09 mg RE/g. However, all bark extracts had lower TFC compared to the leaves extracts. Among these, HAC MeOH exhibited the highest TFC at 10.33 mg RE/g, while MAC water extracts had the lowest flavonoid content at 2.23 mg RE/g.

Phenolic compounds, found in plants, are the primary bioactive components with free radical scavenging and antioxidant capabilities [26]. Thus, polyphenol-rich foods and edibles are deemed health-enhancing [27]. The *Ziziphus* species are both medicinally important and nutritionally rich, serving various beneficial purposes like cooling, stimulating appetite, and aiding digestion [10,11,12]. Numerous studies have concurred with the present investigation’s findings, revealing substantial phenolic and flavonoid content in *Ziziphus*. For example, Uddin et al. conducted a study involving six genotypes of *Z*. *nummularia* and reported TPC values ranging from 82.063 to 88.893 mg GAE/100 g, along with TFC ranging from 63.350 to 76.083 mg QE/100 g in the MeOH extract [28]. Riaz et al. found that drought conditions favor a 3.6% increase in total phenols and a 3.9% increase in flavonoids in *Ziziphus* species compared to irrigated conditions, highlighting the significant impact of growing conditions on phytochemical content [29]. In another prior study, a MeOH extract of *Z. jujuba* exhibited the highest phenolic content (218.33 GAE μg/mg) among various non-polar extracts [27].

### 2.2. Chemical Composition by the Comprehensive Analysis of High-Resolution QTOF and Multiple-Stage Mass Spectrometry in Ion Trap

Identification of phytoconstituents was obtained combining the observation of diode array spectra for the compounds presenting significant UV absorption, high-resolution and MS/MS data from QTOF, as well as multiple-stage mass spectrometry fragmentation pathways (MS^n^) from the Ion Trap. Bark and leaves analysis showed the presence of several classes of secondary metabolites, and different compositions were observed for the two plant parts. The complete characterization is reported in Table 2, where the presence of each of the identified compounds in the different extracts obtained from bark and leaves is indicated.

More in detail, stem bark extract obtained in methanol (Figure 1) present intense signals at high retention time that can be ascribed to lipid fractions, mostly phospholipid derivatives. Several less intense signals can be ascribed to numerous compounds belonging to different classes of phytoconstituents and plant metabolites. A first group of compounds present low retention times and one compound showing molecular formula of C_7_H_13_NO, being tentatively assigned on the basis of the HR-MS and the fragment at *m/z* 71 to pyrrolidine alkaloid named Norhygrine (1-(pyrrolidin-2-yl)propan-2-one) [30]; this alkaloid has been also reported from the bark of the root of *Punica granatum* [31].

The UV spectra of the peaks observed in the stem bark extract mostly were related to flavan-3-ols presenting UV maximum at 280 nm. Procyanidin B2 was identified from the molecular formula C_30_H_27_O_12_ and the diagnostic fragments supporting the reaction of quinone methide fission (leading to *m/z* 289 and 291), retro Diels–Alder fission (leading to *m/z* 427 and 409), and heterocyclic ring fission (leading to *m/z* 453). Furthermore, the identity of compounds was confirmed by co-injection of an authentic standard. Peaks showing similar UV were also identified as flavan-3-ols, as catechin and epicatechin that were identified thanks to the observation in MS2 of *m/z* 139 [18], and also confirmed by standard injection. A derivative presenting molecular formula of C_17_H_18_O_6_ was tentatively assigned as a dimethyl derivative of catechin (or epicatechin). Root extract presents three isobaric peaks that have been tentatively identified as Secoisolariciresinol-sesquilignans derivatives on the basis of their mass spectra [32] but are not detectable in leaves. Phloridzin was also identified on the basis of MS spectrum [33] and confirmed by comparison with reference compounds. Other compounds that were identified both in bark and leaves, and are known constituents of ziziphus, are the isoquinoline alkaloid Magnoflorine [34] and the cyclopeptide Nummularine [35].

Leaves extract presents a higher number of compounds, and several peaks present UV ascribable not only to flavan-3-ols but also flavonols with UV maximum at 350–365 nm. Furthermore, peaks can be ascribed to coumarin derivatives based on the observed UV maximum at 320–330 nm. Compounds were identified from the HR-MS and fragments observed in Ion Trap and confirmed, when possible, with reference compounds. Identified compounds have been luteolin and myricetin-3-O-rhamnoside. Other relevant constituents of the leaves were the flavonoids spinosin, isospinosin, quercetin-3-O-glucoside, quercetin-3-O-rhamnoside, and myricetin-3-O-rhamnoside. All these phenolics and others reported in the table were previously described for ziziphus [18,36]. Further tentatively identified compounds are the pterosupin a C-glycoside phenol derivative that was previously isolated from *Pterocarpum marsupium* [37]. Also identified was the magnoflorine, an aporphyne alkaloid that was previously detected in many other medicinal species [38]. On the other hand, as previously reported in the literature [35,39,40] N-formylcyclopeptide alkaloids were detected, namely the Nummularin F, B, and U, both in bark and leaves. The chromatograms are given in Figure 1 and Figure 2.

As reported in Table 2, several classes of constituents were identified by comparing the obtained data with the literature; some compounds were annotated on the basis of HR-MS data and MS^n^ fragmentation, others also confirmed by injection of reference compounds. Many constituents of *Z. mauritania* were reported in the recent paper by Qin et al. [34], namely the flavonoids, the phenylpropanoids, and the alkaloids, and our findings are in agreement with the previously reported identifications. Furthermore, we also identified the presence of some cyclopeptides, namely the nummularins that were described in *Ziziphus* [41]. The identification of the nummularins was supported by comparison of previously published mass spectrometric data [35]. Norhygrine is a pyrrolidine derivative previously reported from Asian *Sedum* species [30], but we tentatively identify it for the first time in *Z. mauritania*. Catechin and derivatives and procyanidins were detected and mass spectra compared with the literature [42,43] and finally compared with reference compounds as Phloridzin that also was previously identified in *Ziziphus* [33]. Other constituents previously described in this specie were the coumarins [44]. Secolignans were tentatively identified by comparison with mass spectrometric data [32]. Leaves present a larger number of identified constituents, with spinosin, isospinosin, myricetin, and quercetin derivatives only detected in the leaves, while secolignans were only detected in the bark.

The most abundant constituents were also quantified using a semiquantitative approach generating calibration curves with the appropriate reference standard. The most relevant bioactive phytoconstituents that can be extracted from the bark are catechin, epicatechin, and procyanidin B2, while flavonoids were prevalent in the leaves. Data are summarized in Table 3, and results are reported as mg/g of dried extract. An exemplificative chromatogram for the leaves extract is reported in Figure 2. Differences were observed in the extracts obtained with the homogenization (H) and maceration (Ma) with the different solvents. The comparison of extracts obtained with the same solvent but with the two techniques allows to observe a significant effect of the homogenization approach. Considering HAE and MAC for the flavonoids spinosyn, isospinosin, Quercetin-3-O-glucoside, Quercetin-3-O-rhamnoside, and Myricetin-3-O-rhamnoside, for these derivatives we observed a significant increase in the extraction using HAE, with changes of 5–10% compared to MAE. Several compounds present increased extraction using HAE, showing the improved efficiency compared to traditional maceration.

Such comparison also offered the opportunity to compare the extraction in water and with methanol. Aqueous extraction is in general less efficient compared to methanol in the observed dataset. Several phytoconstituents present limited water solubility, thus offering a less favorable solvent for extraction.

The observed differences comparing the homogenization and maceration results on the selected compounds are limited. A general, more efficient extraction using the homogenization approach is observed with average increment of the extraction of compounds in the range 5–10%. This result should be considered also taking into consideration the long time of the extraction for the two approaches. For homogenization, the extraction time was 5 h and for maceration was 24 h, thus a long time for possible extraction and equilibration of the compound’s concentration into and outside from the plant matrix.

### 2.3. Antioxidant Capacity

Aging and various human diseases, including inflammatory disorders, cancer, neurodegenerative conditions, and digestive ailments, have been linked to the excessive production of reactive oxygen species (ROS) and other free radicals [45,46]. Consequently, there has been a growing interest in seeking natural exogenous sources of antioxidants, with plant-based sources being considered a safer alternative to synthetic antioxidants [47]. In pursuit of this objective, previous study investigated the antioxidant efficacy of *Ziziphus* species, employing various strategies for assessing the antioxidant potential of test samples [48].

The current investigation results illustrate a strong correlation between antioxidant activity and TPC in the extracts. For instance, the HAE method using MeOH as the solvent demonstrated significantly higher antioxidant activity across all tested assays compared to MAE with MeOH. The highest antioxidant activity in leaves HAE MeOH extracts was observed in the ABTS assay, with a value of 747.25 mg TE/g, followed by the CUPRAC assay 698.46 mg TE/g, DPPH assay 414.30 mg TE/g, FRAP assay 325.59 mg TE/g, and PDB assay 3.91 mmol TE/g (Table 4). Similarly, the MeOH extracts from the bark displayed higher antioxidant activity, with the highest value recorded in the CUPRAC assay, 741.14 mg TE/g. The findings align with the information presented in Table 1 and Table 2, wherein the MeOH extracts from both leaves and bark exhibited elevated levels of secondary metabolites. Notably, spinosin, isospinosin, and quercetin derivatives were identified as the predominant compounds, with the highest concentration observed in the MeOH extract of leaves. Prior investigations also reported heightened antioxidant activity associated with spinosin [49,50]. Among the water extracts, HAE leaves extract exhibited the highest CUPRAC and FRAP activity, measuring 262.89 mg TE/g and 119.78 mg TE/g, respectively. In contrast, HAE bark extract displayed the highest ABTS and DPPH scavenging activity, measuring 189.52 mg TE/g and 103.25 mg TE/g, respectively. In MAC water extracts, both bark and leaves extracts showed the lowest PDB activity, with values of 0.84 mmol TE/g and 0.81 mmol TE/g, respectively.

A previous study reported that *Z*. *mauritiana* leaf extract has shown effectiveness in alleviating oxidative liver damage, possibly achieved through the scavenging of free radicals by antioxidant enzymes [48]. This plant species is notably abundant in carotenes and various phenolic compounds, including caffeic acid, p-hydroxybenzoic acid, ferulic acid, and p-coumaric acid [51], with naringenin as a predominant flavonoid [52]. Additionally, *Ziziphus mauritiana* fruit exhibits elevated levels of tannin and phytate [53], both of which possess antioxidant properties [54,55]. A study claimed that stressful desert conditions significantly improved the phytochemistry of *Ziziphus* species, leading to increased phenols and flavonoids in leaves and fruits, along with enhanced antioxidant activity [29]. Additionally, a separate investigation revealed that polar solvent extracts exhibited higher antioxidant activity compared to non-polar solvent extracts in *Z*. *jujuba*. Notably, the MeOH extract demonstrated the most potent free radical-scavenging activity, with an IC_50_ value of 20.44 ± 0.18 μg/mL [27]. The MAC activity of the leaves extracted using MeOH was comparable to the MAC activity reported in a previous study on *Z*. *jujuba* fruits (11.04 mg EDTAE/g).

### 2.4. Enzyme Inhibitory Activity

#### 2.4.1. Neuroprotective Effects

Alzheimer’s disease (AD) is a neurodegenerative disorder marked by the progressive loss of memory and cognitive function. The anticipated increase in its frequency among the elderly demographic is expected to increase 3-fold by 2050 [56]. AD resides within the category of conditions that continue to present challenges to medicinal chemists. The quest for highly efficacious drugs to address AD remains an ongoing endeavor. In this context, the inhibition of cholinesterase stands as a promising strategy to elevate acetylcholine levels in the brain [57]. The two inhibitors, AChE and BChE, show immense promise and have attracted considerable attention from researchers.

In the present investigation, the AChE and BChE inhibitory activity was determined. The results indicated in Table 5 showed enzyme inhibitory activity: the HAE MeOH extract from the leaves displayed the highest AChE inhibitory activity, measuring 2.55 mg GALAE/g, followed closely by the bark MAC MeOH extract at 2.41 mg GALAE/g, indicating its potential in neuroprotective effect. In contrast, the HAE water extract from the leaves exhibited the lowest AChE inhibitory activity, measuring only 0.09 mg GALAE/g. Interestingly, there was no inhibition observed in the MAC water extracts from both leaves and bark, as well as in the HAE water extracts from the bark (Table 5).

For BChE inhibition, the MeOH extracts from the bark in both the HAE and MAC methods showed values of 1.57 mg GALAE/g and 1.14 mg GALAE/g, respectively. However, none of the other extracts from both parts of the plant exhibited any BChE inhibitory activity.

The MeOH extracts examined in this study were found to contain abundant spinosin and isospinosin (refer to Table 2), suggesting a potential explanation for the observed AChE) inhibition. Recent research indicates that spinosin may offer benefits in addressing learning and memory deficits associated with AD through its impact on multiple targets [58]. Additionally, Wang et al. reported in their study that a spinosin derivative isolated from *Z*. *mauritiana* demonstrated the inhibition of AChE [59]. Secondary metabolites isolated from *Z*. *oxyphylla* roots inhibited AChE and BChE at different concentrations. *Ziziphus* is rich in bioactive compounds; a study found peptids of *Z*. *jujuba* responsible for the inhibition of AChE and BChE [60].

#### 2.4.2. Dermatoprotective Effects

Tyrosinase plays a pivotal role as the key enzyme in pigment synthesis, initiating a series of reactions that convert the amino acid tyrosine into the melanin biopolymer [61]. Tyrosinase inhibitors hold particular significance in cosmetic applications for their skin-whitening effects [62]. The production of abnormal melanin pigmentation, resulting in a dark complexion, presents a significant aesthetic concern for individuals [63]. Given that plants offer a rich source of bioactive chemicals with a minimal risk of adverse side effects, there has been a growing interest in harnessing them as a natural source of tyrosinase inhibitors. This heightened interest stems from the fact that tyrosinase is a crucial enzyme in the melanin synthesis pathway known as melanogenesis, making it the primary and successful target for inhibitors that directly impede its catalytic activity. It is noteworthy that a majority of commercially available cosmetics and skin-lightening agents are, in fact, tyrosinase inhibitors [64].

In terms of tyrosinase inhibitory activity, the MeOH extracts from both leaves and bark, processed by HAE, demonstrated the highest tyrosinase inhibitory activity levels at 75.97 mg KAE/g and 75.48 mg KAE/g, respectively (Table 5). These results were closely followed by the MAC MeOH extracts from the bark at 74.95 mg KAE/g and from the leaves at 73.79 mg KAE/g, respectively. In contrast, the MAC water extracts from both leaves and bark displayed lower tyrosinase activity when compared to the tested MeOH extracts. Among these, the bark extract showed the lowest tyrosinase activity, measuring only 13.55 mg KAE/g. According to Moon et al., the inhibitory effects of certain flavonoids, including spinosin, quercetin, and kaempferol isolated from *Z*. *jujuba*, were investigated. Among these compounds, spinosin demonstrated the most potent tyrosinase inhibitory activity [65].

#### 2.4.3. Antidiabetic Effect

Hyperglycemia, an excess of blood sugar, is a serious concern. Elevated post-meal glucose levels can predict diabetes complications [66,67]. Most synthetic antidiabetic drugs target type 2 diabetes by addressing insulin issues, but some like metformin have adverse effects at high doses [68]. Thus, the main goal in antidiabetic research is to find safe and effective agents without side effects. In pursuit of this objective, researchers have turned to traditional medicines and foods from diverse cultures, exploring them for potential clues and insights that could lead to the development of new therapeutic drugs [26,69,70]. Targeting enzymes like α-amylase and α-glucosidase can help reduce post-meal hyperglycemia [71,72].

Table 4 shows inhibition of glucose digestive enzymes. The MeOH extracts demonstrated the highest α-amylase inhibitory activity. The HAE-MeOH extract from the leaves exhibited an inhibition value of 0.78 mmol ACAE/g, followed closely by the MAC-MeOH extract from the leaves at 0.76 mmol ACAE/g, and the HAE MeOH extract from the bark at 0.72 mmol ACAE/g, suggesting its potential in modulating cholinergic neurotransmission [73]. The MAC-MeOH extract from the bark showed an inhibition value of 0.69 mmol ACAE/g. On the other hand, the water extracts from the leaves, obtained through MAC, exhibited a lower inhibition value of 0.15 mmol ACAE/g, while the water extracts from the bark, obtained through HAE, had a lower inhibition value of 0.13 mmol ACAE/g. In terms of α-glucosidase inhibitory activity, the MeOH extracts from the leaves, both from MAC and HAE, exhibited higher and similar levels of inhibition, measuring 2.11 mmol ACAE/g. However, their bark extracts showed no inhibition. The water extracts from the bark, obtained through both HAE and MAC, exhibited similar inhibition values of 1.87 mmol ACAE/g.

Our discovery regarding the MeOH extracts from *Ziziphus* exhibiting inhibition against α-amylase and α-glucosidase aligns with prior research findings [74,75]. Suksamrarne and colleagues demonstrated in their study that the MeOH extract of various *Ziziphus* plants displayed superior α-glucosidase inhibitory activity compared to their other extracts [74].

### 2.5. Multivariate Data Analysis

In this work, we generated large amounts of experimental data as results of spectrophotometric assays, enzymatic inhibitions. Thus, to establish correlations between observed bioactivities and chemical constituents, we adopted multivariate data analysis using PCA and OPLS-DA. We can observe that the extracts on the basis of the spectrophotometric and enzyme inhibitory assays can be divided into three similar groups, one that is in the –x; +y part of the plot is formed by the Leaves HAE methanol extract and this group presents TFC and glucosidase activities as the most discriminant descriptors. On the opposite side (+x; +y), the group was formed by bark HAE and MAC with methanol, which strongly correlates with butyrlcholinesterase inhibition. On the lower side of the plot (–y) is a group formed by leaves HAE water and bark HAE water that is more correlated with MCA. In the end, the obtained data can help in the description of the observed activities and spectrophotometric determination obtained for the extracts. Namely, methanol solvent, when used both with the maceration or the HAE, is strongly associated with good results in almost all the antioxidant assays, namely TRAP, TPC, DPPH, CUPRAC, and ABTS. The most relevant activities on the observed enzymes are the tyrosinase and amylase and acetylcholinesterase. This information shows that the extraction with methanol from the leaves and the bark is able to yield compounds with high antioxidant capacity and significant inhibitory activities. At the same time, a different behavior is observed considering leaves and bark. In fact, the leaves are strongly correlated also with TPC and glucosidase, while bark mostly with butyrlcholinesterase inhibition. On the other hand, the samples extracted with water are more related to MCA activity. All these results are summarized in Figure 3 that presents the superimposition of the loading plot of the OPLS-DA, and in the red square, the corresponding score plot showing the most relevant variables for the group of extracts.

Considering the results of the chemical analysis, we can observe that the leaves are rich in spinosyn and isopsinosin compared to the bark and present quercetin-3-O-rhamnoside as well as luteolin. Furthermore, only the leaves contain magnoflorin, caftaric acid, and gallocatechin, and all these compounds may present some specific activities on the observed enzymes. Bark is, on the other hand, characterized by the presence of some secoisolariciresionl-sesquilignan derivatives that are not detected in the leaves.

### 2.6. Molecular Docking

Molecular docking was used to examine how the main components in the *Z. mauritiana* extracts bound to the selected target enzymes. Figure 4 displays the ligands’ binding energy scores. Spinosin, isospinosin, and quercetin-3-*O*-glucoside were predicted to have the highest binding energy against the target enzymes. For instance, spinosin strongly bound to the active site of AChE mainly via H-bonding Tyr72, Asp74, Asn283, Phe295, and Arg296; as well as π–π stacked and π–π T-shaped with Trp286 and Tyr341, respectively (Figure 5A). Similarly, the same compound formed firm interactions with BChE mainly via H-bonding with Thr120, Tyr128, and His438 deep inside the active site supported by the π–π T-shaped and amide–π stacked interactions close to the entrance to the channel (Figure 6B). Interestingly, the key interactions between the built human tyrosinase model and isospinosin were H-bonds with Gln359, Asn364, and Gln378. Other supporting interactions, like in the above complexes, were multiple van der Waals interactions, including those with the catalytically essential active site cupper ions (Figure 6A). Furthermore, spinosin was accommodated in the catalytic cavity of amylase via the formation of H-bonds with Thr163, Gly304, and Asp356, as well as π–π T-shaped interactions with Trp59 and His304 (Figure 6B). Finally, the key interactions between the model of human glucosidase and spinosin were also H-bonds with Asp168, Arg173, Glu239, Arg267, strengthened by hydrophobic contacts with Ala40, Ala65, and Ile66, as well as multiple van der Waals interactions all over the active site (Figure 6C). The dominant compounds in the extracts of *Ziziphus mauritiana* bark and leaves extracts may therefore be interacting with the chosen proteins to inhibit their functions.

## 3. Materials and Methods

### 3.1. Plant Material

In the summer of 2021, the leaves (almost 500 g) and barks (almost 200 g) of *Z. mauritiana* were gathered in the vicinity of Bouake (Côte d’Ivoire) by the botanist Ouattara Katinan Etienne. Voucher specimens were deposited at the herbarium in the Nangui Abrogoua University (KIS-21-175). Prior to extraction, the plant materials (leaves and stem barks) were carefully washed with tap and distilled water to eliminate any soil and contaminants. After being air-dried for 10 days (in shade at room temperature), the plant samples were powdered.

### 3.2. Sample Preparation

Two extraction techniques were used to prepare the extracts, namely homogenizer-assisted (HAE) and maceration (MAC). Two solvents (methanol and water) were used in these techniques. To prepare HAE, 5 g of plant samples were extracted with the solvents (100 mL) at 6000 g in a homogenizer for 5 h (IKA-Ultraturrax). Regarding MAC, the plant samples (5 g) were macerated with the solvents (100 mL) for 24 h at room temperature. After extraction, all extracts were filtered, and the methanol extracts were dried using a rotary evaporator. As for the water extracts, after filtration, the extracts were dried under vacuum using a lyophilizer. All extracts were stored at 4 °C until analysis.

### 3.3. Assay for Total Phenolic and Flavonoid Contents

According to the methods specified by [76], total phenolics and flavonoids were quantified. Gallic acid (GA) and rutin (RE) served as standards in the assays, and the outcomes were reported as gallic acid equivalents (GAEs) and rutin equivalents. All experimental details are given in the Appendix A.

### 3.4. High-Resolution LC-QTOF-MS and Multiple-Stage Mass Spectrometry by Ion Trap Analysis

The phytochemical analysis was performed combining ultraperformance liquid chromatography with diode array and electrospray Quadrupole Time Of Flight Mass Spectrometry (Agilent Technology, Santa Clara, CA, USA) and Multiple-Stage fragmentation in Ion Trap (Agilent Technology). An Agilent 1290 UPLC system equipped with an autosampler and 1290 series Diode Array was used as chromatograph. After the column, the flow was split with a passive T junction, and liquid was sent to diode array or mass spectrometer. The Agilent 6530 QTOF (Agilent Technology) was used as a detector; the instrument is equipped with a Jet Stream source and was operating in positive ion mode. During the acquisition, the mass values were calibrated using the Agilent calibration mixture. The parameters of the MS were as follows: gas temperature 350 °C, fragmentor 250 V, skimmer 45 V, drying gas 6 L/min, nebulizer 25 psig, Shealth temp 275 °C.

As stationary phase, an Agilent SB C18 (3 × 100 mm; 1.7 micron) was used. Eluents were water 0.1% formic acid (A), and acetonitrile with 0.1% formic acid (B). The flow rate was set to 0.5 mL/min. Gradient started with 90% of A remaining isocratical for 1 min then going to 85% B in 10 min, and then reached 90% B in 12 min and remained isocratical up to 14 min. For the multiple-stage mass spectrometry, an Agilent 1260 chromatograph was used. As stationary phase, an Agilent SB C-18 (Agilent Technology) was used (4.6 × 50; 1.8 micron). As eluents, water 1% formic acid (A), Acetonitrile (B), and methanol (C) were used. Gradient started with 95% A and 5% B, then in 2.5 min 85% A and 15% B, then in 12 min 80% A, 18% B and 2% methanol, and in 15.5 min 50% A, 40% B, and 10%C, then at 19 min 20% A, 70% B, and 10% C. The flow rate was 0.750 μL/min. Flow after the column was split by a T passive junction, and half was sent to diode array and half to a Varian MS500 (Varian Technology, Ontario, ON, Canada) equipped with an ESI source that was used and operated in turbo DDS mode acquiring MS^n^ spectra of eluted species. Drying gas was set at 45 psi, while nebulizer at 18 psi. The drying gas temperature was 340 °C at the beginning of the chromatography and decreased at 5 °C/min for 20 min. Capillary was set to 80 °C, RF loading was 85%. For semiquantitative analysis, reference compounds were used selecting compounds belonging to flavan-3-ols, flavonoids, and coumarins. Calibration curves for catechin, epicatechin, gallocatechin, and procyanidin B2 were obtained using solution at four different concentrations of the four reference standards, at 2 µg/mL, 10 µg/mL, 20 µg/mL, and 50 µg/mL, and collecting chromatogram at 280 nm. Quercetin-3-O-glucoside was used for quantification of the flavonoids, and a calibration curve was obtained using solutions obtained at 2 µg/mL, 10 µg/mL, 20 µg/mL, and 50 µg/mL and collecting the chromatogram at 350 nm. For the coumarin quantification, auraptene and geranyl umbelliferone were used, generating calibration curves using solutions at 5 µg/mL, 10 µg/mL, 15 µg/mL, and 30 µg/mL and collecting the chromatogram at 330 nm. An exemplificative chromatogram of leaves extract is reported in Figure 3.

Data obtained from the QTOF (high-resolution) and Ion Trap (low-resolution) mass spectrometry were combined to establish the identity of the eluted compounds.

### 3.5. In Vitro Bioactivity Assays

The assessment of the biological activity of the obtained extracts was carried out by measuring the antioxidant [77], and enzyme-inhibitory potential [77]. With the aim of comprehensive insight into the real antioxidant potential, six different tests were used (DPPH, ABTS, FRAP, CUPRAC, PBD, and MCA tests). The enzyme-inhibitory potential of obtained extracts was estimated towards five different clinically important enzymes: tyrosinase, amylase, glucosidase, acetylcholinesterase, and butrylcholinesterase. All experimental details are given in the Appendix A.

### 3.6. Molecular Modeling

The X-ray crystal structures of the following target proteins were downloaded from the protein data bank (https://www.rcsb.org/ accessed on 20 May 2024) [78]: AChE (PDB ID: 6O52) [79] α-amylase (PDB ID: 1B2Y) [80], BChE (PDB ID: 6EQP) [81]. A homology model of human tyrosinase constructed using UniProt entry P14679 as the target sequence and using the crystal structure of tyrosinase from *Priestia megaterium* (PDB ID: 6QXD) [82] as the template was retrieved [83]. Moreover, the homology model of human α-glucosidase built using the crystal structure of *Mus musculus* glucosidase (PDB ID: 7KBJ) [84] as a template, and using the UniProt entry P0DUB6 as the target sequence, was obtained from the same study [83]. Protein preparation was performed according to the protocol described previously [85]. Each ligand’s three-dimensional structure was retrieved from the ChemSpider database (https://www.chemspider.com/ accessed on 20 May 2024). The UCSF Chimera tool was used to optimize the geometry of the ligand’s three-dimensional structures [86]. With the help of MGLTools 1.5.6 software, all docking grid files were created utilizing the cocrystal ligand in each crystal structure. This program combines all hydrogen atoms and gives all protein atoms Gasteiger partial charges. The π Lamarckian genetic method included in AutoDock 4.2.6 software was used to simulate docking (https://autodock.scripps.edu/ accessed on 20 May 2024) [87], implementing the docking protocol as detailed in Ref. [88]. Using Biovia DS Visualizer v4.5, each ligand’s binding energy scores were calculated, and protein–ligand interactions were investigated (BIOVIA, San Diego, CA, USA).

### 3.7. Statistical Analysis

Statistical analysis was performed using Xl Stat (Version 16). All analyses were conducted in triplicates (n = 3) and presented as mean values with their standard deviation (mean value ± std). Differences between samples were examined using one-way analysis of variance (AVOVA) and Tukey’s post hoc test with significance level set at *p* < 0.05. For multivariate analysis, the data from the spectrophotometrical assays and enzyme inhibitory assays were used to generate a table containing the extracts and the respective results. The matrix was loaded in SIMCA 12 and pareto-scaled. Data initially were used to obtain a PCA, then an OPLS-DA to describe the different obtained extracts.

## 4. Conclusions

The comparison of extraction methods for *Z. mauritiana* bark and leaves showed that homogenization-assisted extraction (HAE) using MeOH was more effective, resulting in higher phenolic content and antioxidant activity compared to maceration extraction (MAE) with MeOH. It is important to note that the MeOH and water extracts from the leaves contained higher levels of secondary metabolites, such as spinosin and isospinosin. Based on the chemical structures of the most abundant compounds, we can infer that the flavonols present contributed to the antioxidant and enzyme-inhibitory activities observed in the extracts. In terms of industrial insights, our results can be valuable, and we have suggested that the HAE technique can be more effective in producing functional applications, including novel pharmaceuticals or cosmeceuticals, by using *Z. mauritiana*. In future studies, the main components of *Z. mauritiana* extracts will be isolated, with the aim of investigating in more detail the role of the different classes of phytoconstituents in the observed bioactivities. However, researchers should consider the toxicity of methanol, and we suggest the use of ethanol or an ethanol/water mixture in future studies.

## Figures and Tables

**Figure 1 plants-13-02195-f001:**
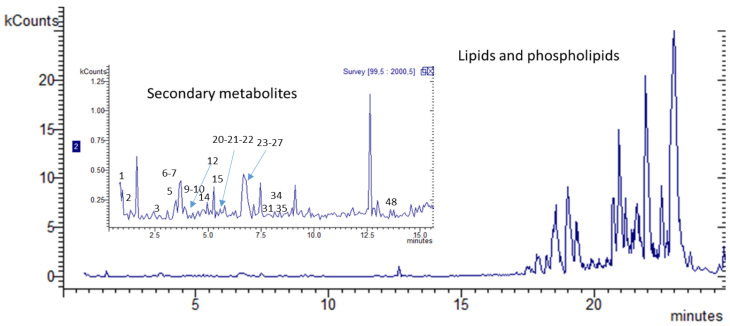
The methanol extract of stem bark, BPI chromatogram Ion Trap detector, negative ion mode.

**Figure 2 plants-13-02195-f002:**
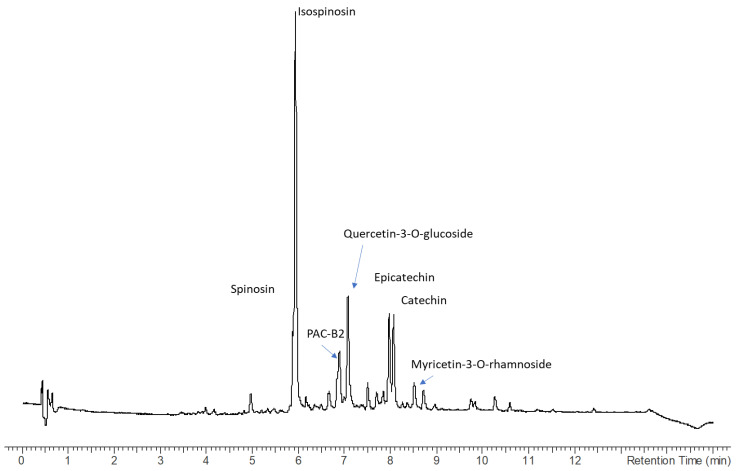
The methanol extract of leaves observed at 254 nm used for the semiquantification of compounds, main derivatives are highlighted.

**Figure 3 plants-13-02195-f003:**
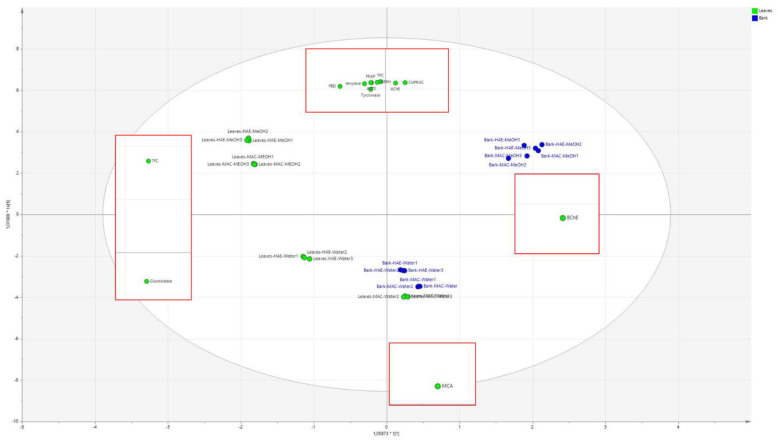
Combination of the loading plot of the OPLS-DA of the different obtained extracts of ziziphus leaves and barks and the corresponding score plot that represents the results of the spectrophotometric assays and enzyme inhibitory results.

**Figure 4 plants-13-02195-f004:**
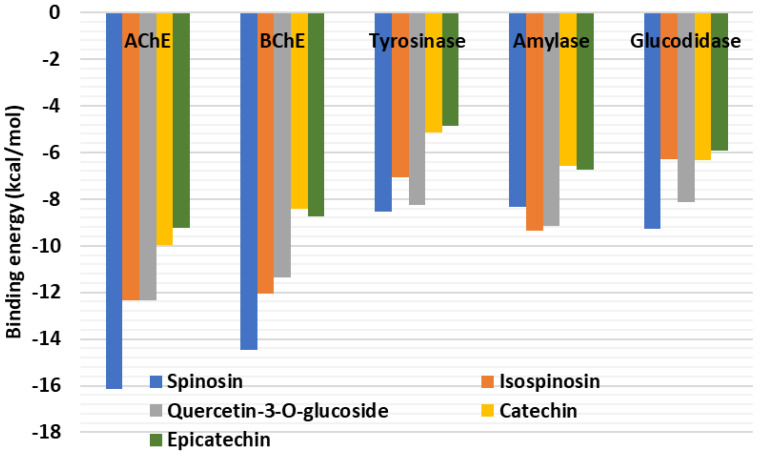
Docking score of dominant compounds in the extracts of *Ziziphus mauritiana* bark and leaves extracts.

**Figure 5 plants-13-02195-f005:**
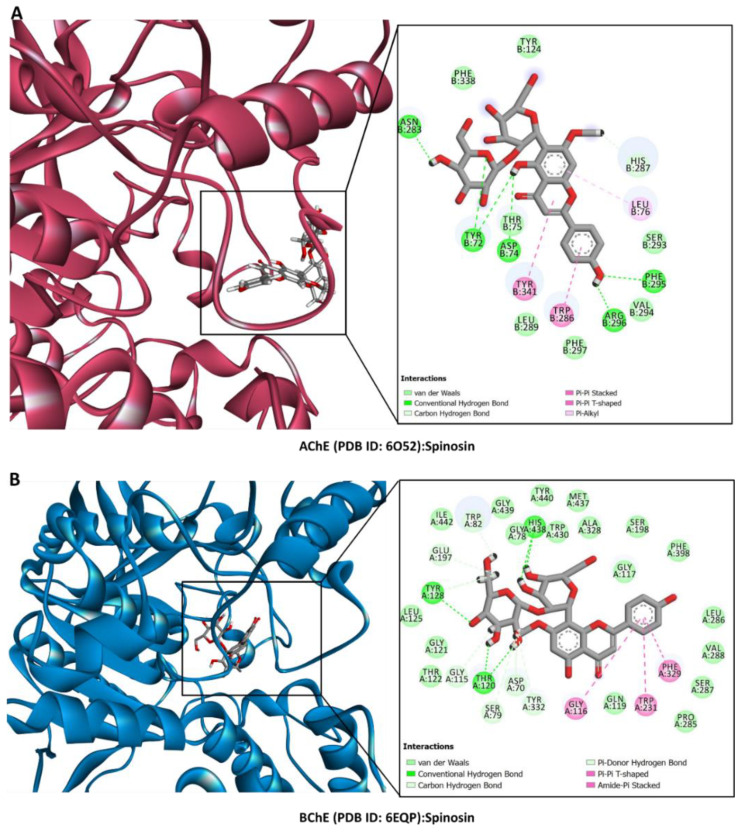
Protein–ligand interaction: (**A**) AChE and spinosin, (**B**) BChE and spinosin.

**Figure 6 plants-13-02195-f006:**
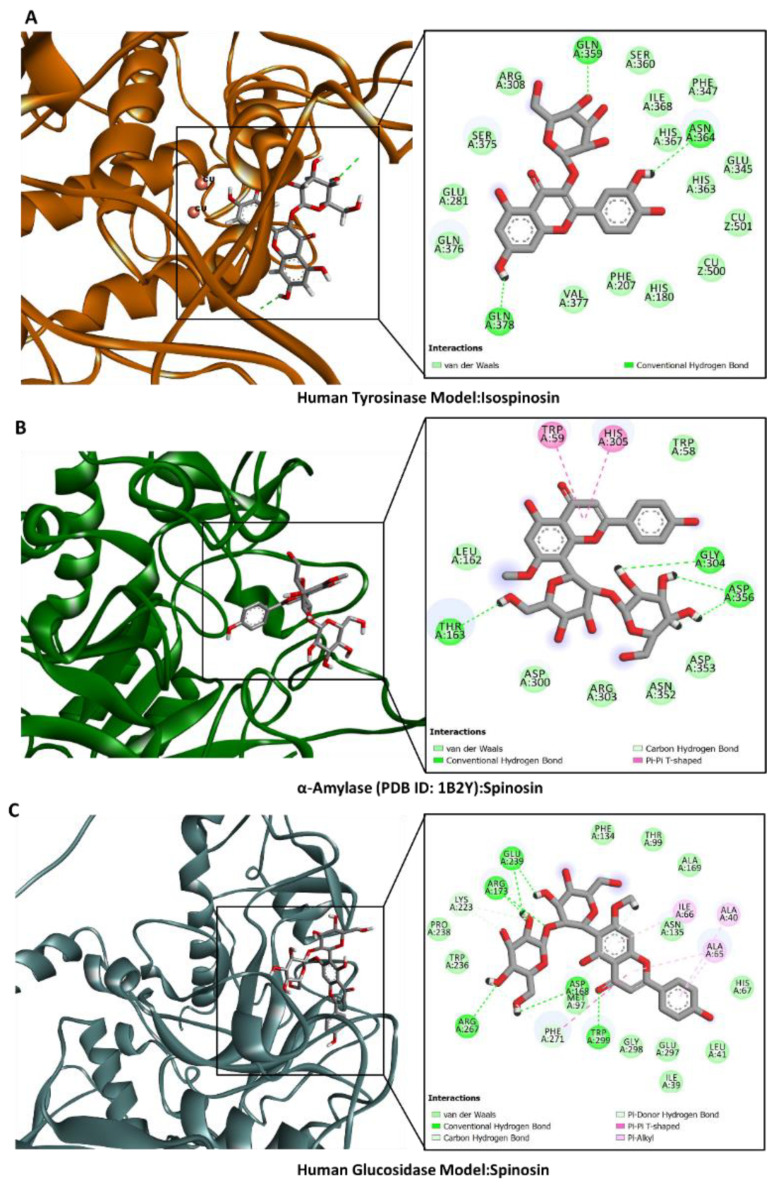
Protein–ligand interaction: (**A**) tyrosinase and isospinosin, (**B**) amylase and spinosin, and (**C**) glucosidase and spinosin.

**Table 1 plants-13-02195-t001:** Total phenolic and flavonoid contents in the tested extracts.

Parts	Methods	Solvent	TPC (mg GAE/g)	TFC (mg RE/g)
Leaves	HAE	MeOH	112.01 ± 1.08 ^a^	51.91 ± 0.15 ^a^
	MAC	MeOH	100.24 ± 0.86 ^c^	48.09 ± 0.20 ^b^
	HAE	Water	49.26 ± 0.10 ^d^	20.80 ± 0.45 ^c^
	MAC	Water	34.57 ± 0.53 ^g^	11.52 ± 0.40 ^c^
Bark	HAE	MeOH	105.99 ± 0.92 ^b^	10.33 ± 0.06 ^d^
	MAC	MeOH	101.02 ± 1.20 ^c^	6.95 ± 0.09 ^e^
	HAE	Water	45.48 ± 0.57 ^e^	4.35 ± 0.06 ^f^
	MAC	Water	42.78 ± 0.28 ^f^	2.23 ± 0.32 ^g^

Values are reported as mean ± SD of three parallel experiments. HAE: Homogenizer-assisted extraction; MAC: Maceration; TPC: Total phenolic content; TFC: Total flavonoid content. GAE: Gallic acid equivalent; RE: Rutin equivalent. Different letters indicate significant differences in the tested extracts (*p* < 0.05).

**Table 2 plants-13-02195-t002:** Chemical characterization of the tested extracts: the symbol “x” indicates the presence of the compound in the extract, * indicates confirmation with comparison of authentic standard. Quantification of the most abundant constituents was performed as described in the Materials and Methods.

No.	Rt	MS	Formula	Name	Stem Bark Methanol	Stem Bark Water	Leaves Methanol	Leaves Water
1	0.56	173.0806	C8H12O4	octene dioic acid	x	x	x	x
2	0.65	127.0752	C7H13NO	Norhygrine	x	x	x	x
3	2.65	289.1762	C10H20N6O4	Asparagylarginin	x	x	x	x
4	3.35	609.1811	C28H33O15	Spinosin			x	x
5	3.46	121.0508	C7H7NO	Benzamide	x	x	x	x
6	3.63	609.1811	C28H33O15	Isospinosin		x	x	x
7	3.81	579.1952	C30H27O12	Procyanidin B2 *	x	x	x	x
8	3.9	465.1833	C21H21O12	Quercetin-3-O-glucoside *			x	x
9	4.03	429.152	C23H32N4O4	Nummularine F	x	x	x	x
10	4.08	319.117	C17H18O6	dimethyl catechin	x	x	x	x
11	4.17	449.1952	C21H21O11	Quercetin-3-O-rhamnoside *			x	x
12	4.29	593.123	C32H41N5O6	Nummularine B	x	x	x	x
13	4.44	433.1402	C18H24O12	Apiosylglucosyl 4-hydroxybenzoate			x	x
14	4.81	291.243	C15H15O6	Catechin *	x	x	x	x
15	5.19	291.243	C15H15O6	Epicatechin *	x	x	x	x
16	5.22	465.139	C21H21O12	Myricetin-3-O-rhamnoside *			x	x
17	5.33	495.1501	C23H26O12	Pimentol	x		x	x
18	5.47	337.198	C12H24N4O7	Fructopyranosil arginine	x		x	x
19	5.93	489.2703	C24H40O10	alpha-Ionol O-[arabinosyl-(1->6)-glucoside]			x	x
20	6.35	559.1371	C30H38O10	Secoisolariciresinol-sesquilignan	x	x		
21	6.48	471.2995	C26H39N4O4	Nummularine U	x	x	x	x
22	6.61	559.1371	C30H38O10	Secoisolariciresinol-sesquilignan	x	x		
23	6.89	559.1371	C30H38O10	Secoisolariciresinol-sesquilignan	x	x		
24	7.04	158.1545	C11H11N	dimethylquinoline	x		x	x
25	7.07	313.0702	C13H12O9	Caftaric acid *			x	x
26	7.09	279.0953	C10H18N2O5S	Methionyl glutamate			x	x
27	7.35	343.1978	C20H24NO4	Magnoflorine	x	x	x	x
28	7.5	287.056	C15H10O6	Luteolin *			x	x
29	7.63	465.0923	C21H21O13	quercetin-hexoside			x	x
30	7.97	327.0421	C15H18O8	Coumaroyl hexoside			X	x
31	8.06	469.3521	C31H48O3	Methyl 3-oxo-12-oleanen-28-oate	x	x	X	x
32	8.26	559.2542	C30H38O10	Secoisolariciresinol-sesquilignan	x			
33	8.26	595.189	C27H31O15	Quercetin-dirhamnoside			X	x
34	8.36	307.0821	C15H15O7	Gallocatechin *	x	x	x	x
35	8.96	437.1445	C21H24O10	Phloridzin *	x	x	X	x
36	9.21	337,1062	C16H16O8	Caffeoyl shikimic acid			x	x
37	9.74	609.2123	C28H33O15	Isorhamnetin-diglucoside			x	x
38	9.83	315.1792	C16H26O6	Carveol hexoside			x	x
39	9.86	559.2542	C30H38O10	Secoisolariciresinol-sesquilignan	x			
40	10.1	327.0421	C15H18O8	Coumaroyl hexoside			x	x
41	11.17	301.141	C14H20O7	Methoxy-benzyl-hexoside	x		x	x
42	11.18	149.0238	C5H8O3S	2-oxo-4-methylthiobutanoate			x	x
43	11.51	299.1839	C19H22O3	Auraptene			x	x
44	12.41	801.598	C44H81O10P	PG(18:3(9Z,12Z,15Z)/20:0 Phosphatidyl glycerol	x			
45	13.5	771.465	C45H89NO8	Phospholipid	x		x	x
46	13.6	850.6961	C50H92NO7P	PC(o-22:1(13Z)/20:4(8Z,11Z,14Z,17Z))	x			
47	13.61	299.1839	C19H22O3	Geranyl umbelliferone			x	x
48	13.8	329.111	C18H16O6	7,8,4′-Trimethylisoscutellarein	x	x	x	x

**Table 3 plants-13-02195-t003:** Measured amounts of selected phytochemicals in the extracts. Values are reported as mean ± SD of three parallel experiments. HAE: homogenizer-assisted extraction; MAC: maceration. Different letters indicate statistical significance comparing the same solvent and the two different techniques (*p* < 0.05).

Name	Stem Bark Methanol HAE (mg/g)	Stem Bark Methanol MAC (mg/g)	Stem Bark Water HAE (mg/g)	Stem Bark Water MAC (mg/g)	Leaves Methanol HAE (mg/g)	Leaves Methanol MAC (mg/g)	Leaves Water HAE (mg/g)	Leaves Water MAC (mg/g)
Spinosin					8.42 ± 0.25 ^a^	7.88 ^b^ ± 0.50	7.25 ± 0.20 ^c^	7.02 ^d^ ± 0.30
Isospinosin			0.25 ± 0.22	0.21 ± 0.12	6.85 ± 0.22 ^a^	6.35 ± 0.22 ^b^	6.19 ± 0.11 ^c^	5.89 ± 0.32 ^d^
Procyanidin B2 *	1.22 ± 0.04 ^a^	1.04 ± 0.06 ^b^	0.54 ± 0.03	0.48 ± 0.13	0.68 ± 0.13 ^c^	0.53 ± 0.10 ^d^	0.34 ± 0.05	0.31 ± 0.05
Quercetin-3-O-glucoside *					6.29 ± 0.20 ^a^	6.01 ± 0.20 ^b^	4.13 ± 0.22 ^a^	3.99 ± 0.12 ^b^
dimethyl catechin	0.28 ± 0.03	0.29 ± 0.03	0.13 ± 0.01	0.15 ± 0.01	0.31 ± 0.13	0.25 ± 0.03	0.26 ± 0.01	0.22 ± 0.01
Quercetin-3-O-rhamnoside					1.46 ± 0.10 ^a^	1.10 ± 0.21 ^b^	1.41 ± 0.19 ^a^	0.99 ± 0.14 ^b^
Catechin *	2.51 ± 0.18 ^a^	2.23 ± 0.18 ^b^	1.41 ± 0.11 ^c^	1.24 ± 0.09 ^d^	6.89 ± 0.15 ^c^	6.35 ± 0.15 ^d^	5.71 ± 0.15 ^e^	5.16 ± 0.19 ^f^
Epicatechin *	3.62 ± 0.16 ^a^	3.19 ± 0.16 ^b^	1.58 ± 0.15 ^c^	1.30 ± 0.22 ^d^	6.72 ± 0.15 ^e^	6.20 ± 0.15 ^f^	4.77 ± 0.22 ^g^	4.19 ± 0.22 ^h^
Myricetin-3-O-rhamnoside					3.25 ± 0.15 ^a^	3.11 ± 0.08 ^b^	2.85 ± 0.12 ^c^	2.35 ± 0.12 ^d^
Luteolin *					0.34 ± 0.02	0.32 ± 0.02	0.12 ± 0.05	0.10 ± 0.05
Gallocatechin	1.39 ± 0.11 ^a^	1.21 ± 0.10 ^b^	0.78 ± 0.10	0.64 ± 0.10	0.49 ± 0.05 ^c^	0.39 ± 0.02 ^d^	0.36 ± 0.05	0.31 ± 0.05
Auraptene					0.21 ± 0.05 ^a^	0.20 ± 0.09 ^a^	0.13 ± 0.03 ^b^	0.16 ± 0.06 ^b^
Geranyl umbelliferone					0.13 ± 0.05	0.10 ± 0.05	0.08 ± 0.05	0.06 ± 0.05
7,8,4′-Trimethylisoscutellarein	0.15 ± 0.02	0.15 ± 0.02	0.07 ± 0.02	0.07 ± 0.02	0.34 ± 0.08	0.28 ± 0.07	0.07 ± 0.02	0.06 ± 0.02

* Confirmed by standards.

**Table 4 plants-13-02195-t004:** Antioxidant effects of the tested extracts.

Parts	Methods	Solvent	DPPH (mg TE/g)	ABTS (mg TE/g)	CUPRAC (mg TE/g)	FRAP (mg TE/g)	PBD (mmol TE/g)	MCA (mg EDTAE/g)
Leaves	HAE	MeOH	414.30 ± 4.73 ^a^	747.25 ± 5.34 ^a^	698.46 ± 4.48 ^b^	325.59 ± 2.99 ^a^	3.91 ± 0.25 ^a^	10.17 ± 0.62 ^d^
	MAC	MeOH	331.83 ± 4.49 ^d^	545.97 ± 5.01 ^d^	693.98 ± 2.92 ^b^	265.05 ± 3.19 ^c^	3.16 ± 0.10 ^b^	11.79 ± 0.64 ^d^
	HAE	Water	92.32 ± 9.99 ^e^	226.51 ± 3.67 ^e^	262.89 ± 2.47 ^c^	119.78 ± 3.07 ^d^	1.29 ± 0.11 ^d^	15.18 ± 0.61 ^c^
	MAC	Water	33.90 ± 1.29 ^f^	67.96 ± 1.65 ^g^	102.33 ± 0.50 ^e^	36.00 ± 0.44 ^f^	0.81 ± 0.04 ^e^	31.71 ± 0.72 ^b^
Bark	HAE	MeOH	365.67 ± 3.74 ^b^	630.63 ± 6.87 ^b^	761.14 ± 22.58 ^a^	279.64 ± 4.44 ^b^	3.11 ± 0.25 ^bc^	14.60 ± 0.72 ^c^
	MAC	MeOH	348.11 ± 1.53 ^c^	601.00 ± 9.43 ^c^	747.38 ± 5.34 ^a^	277.69 ± 5.74 ^b^	2.67 ± 0.20 ^c^	14.90 ± 1.35 ^c^
	HAE	Water	103.25 ± 6.25 ^e^	189.52 ± 11.48 ^f^	259.43 ± 1.67 ^c^	114.96 ± 1.47 ^d^	1.07 ± 0.07 ^de^	30.71 ± 0.10 ^b^
	MAC	Water	13.92 ± 1.43 ^g^	65.58 ± 4.19 ^g^	203.86 ± 3.60 ^d^	88.11 ± 0.91 ^e^	0.84 ± 0.02 ^e^	35.23 ± 0.27 ^a^

Values are reported as mean ± SD of three parallel experiments. TE: Trolox equivalent; EDTAE: EDTA equivalent. Different letters indicate significant differences in the tested extracts (*p* < 0.05).

**Table 5 plants-13-02195-t005:** Enzyme inhibitory effects of the tested extracts.

Parts	Methods	Solvent	AChE (mg GALAE/g)	BChE (mg GALAE/g)	Tyrosinase (mg KAE/g)	Amylase (mmol ACAE/g)	Glucosidase (mmol ACAE/g)
Leaves	HAE	MeOH	2.55 ± 0.02 ^a^	Na	75.97 ± 0.73 ^a^	0.78 ± 0.01 ^a^	2.07 ± 0.01 ^b^
	MAC	MeOH	2.14 ± 0.08 ^c^	Na	73.79 ± 0.44 ^a^	0.76 ± 0.01 ^a^	2.11 ± 0.01 ^a^
	HAE	Water	0.09 ± 0.01 ^d^	Na	47.53 ± 2.08 ^b^	0.25 ± 0.01 ^d^	2.11 ± 0.01 ^a^
	MAC	Water	Na	Na	33.47 ± 0.30 ^c^	0.15 ± 0.01 ^e^	0.20 ± 0.02 ^d^
Bark	HAE	MeOH	2.39 ± 0.06 ^b^	1.57 ± 0.13 ^a^	75.48 ± 0.08 ^a^	0.72 ± 0.02 ^b^	na
	MAC	MeOH	2.41 ± 0.02 ^b^	1.14 ± 0.36 ^b^	74.95 ± 0.74 ^a^	0.69 ± 0.02 ^c^	na
	HAE	Water	Na	Na	28.56 ± 3.02 ^d^	0.13 ± 0.01 ^e^	1.87 ± 0.01 ^c^
	MAC	Water	Na	Na	13.55 ± 0.42 ^e^	0.16 ± 0.01 ^e^	1.87 ± 0.01 ^c^

Values are reported as mean ± SD of three parallel experiments. GALAE: Galanthamine equivalent; KAE: Kojic acid equivalent; ACAE: Acarbose equivalent; na: not active. Different letters indicate significant differences in the tested extracts (*p* < 0.05).

## Data Availability

Data are contained within the article and Appendix A.

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
