# Peer review of "Ziziphus mauritiana Lam. Bark and Leaves: Extraction, Phytochemical Composition, In Vitro Bioassays and In Silico Studies"

_plants, 2024, doi:10.3390/plants13162195_

Round 1

Reviewer 1 Report

Comments and Suggestions for Authors

Main notes:

1.      The title of the paper ‘Ziziphus mauritiana Lam. bark and leaves: Homogeneizer-assisted extraction phytochemical composition, in vitro bioassays in silico studies' could be changed – for instance, into ‘Ziziphus mauritiana Lam. bark and leaves: extraction, phytochemical composition, in vitro bioassays and in silico studies.

2.      In the introduction, it is worth indicating to which botanical family the plant belongs (Rhamnaceae) and where it grows as wild and where as an introduced species https://powo.science.kew.org/taxon/urn:lsid:ipni.org:names:719349-1. It is also worth taking information from this site about the scientist who described this species for science from this resource (Lam.).

3.      It needs to improve the aim and rationale of the study at the end of the Introduction section (lines 72-82). It is worth making this paragraph more concise.

4.      The abbreviations cannot be used in the Abstract without deciphering: line 24 - AChE and in line 375 - Full word ‘acetylcholinesterase’. The same with butyrlcholinesterase: line 24 – BchE and in line 369 - Full word ‘butyrlcholinesterase’.

5.      Lines 85-89The 'Results' section cannot start from the sentences like these ‘Phenolic compounds, found in plants, are the primary bioactive components with...'. It should start with the presentation of authors’ research results, and after it present their discussion and comparison with data from scientific sources.

6.      Figure 3: There is no X-axis on the graph.

7. A moderate check is required for the English language and style through the text. For instance, line 195: While scolignans, were only detected in bark’. The conclusion section should be more clearly formulated, for instance: In our study on Z. mauritiana, we found that homogenization-assisted extraction (HAE) with MeOH was more effective than maceration (MAE) with MeOH in terms of yielding higher phenolic content and antioxidant activity. We also discovered that MeOH and water extracts from Z. mauritiana leaves contain elevated levels of secondary metabolites such as spinosin and isospinosin … ‘

8.      Lines 474-480: six different tests were used for studying antioxidant activity but their methodologies were not described.

Comments on the Quality of English Language

7.      A moderate check is required for the English language and style through the text. For instance, line 195: While scolignans, were only detected in bark’. The conclusion section should be more clearly formulated, for instance: In our study on Z. mauritiana, we found that homogenization-assisted extraction (HAE) with MeOH was more effective than maceration (MAE) with MeOH in terms of yielding higher phenolic content and antioxidant activity. We also discovered that MeOH and water extracts from Z. mauritiana leaves contain elevated levels of secondary metabolites such as spinosin and isospinosin … ‘

Author Response

Reviewer 1: Corrections are indicated in yellow color

  1. The title of the paper ‘Ziziphus mauritianaLam. bark and leaves: Homogeneizer-assisted extraction phytochemical composition, in vitro bioassays in silico studies' could be changed – for instance, into ‘Ziziphus mauritiana Lam. bark and leaves: extraction, phytochemical composition, in vitro bioassays and in silico studies’.

Response: We have changed the title based on the reviewer comments.

  1. In the introduction, it is worth indicating to which botanical family the plant belongs (Rhamnaceaeand where it grows as wild and where as an introduced species https://powo.science.kew.org/taxon/urn:lsid:ipni.org:names:719349-1. It is also worth taking information from this site about the scientist who described this species for science from this resource (Lam.).

Response: We have added more details for the tested plant species.

  1. It needs to improve the aim and rationale of the study at the end of the Introduction section (lines 72-82). It is worth making this paragraph more concise.

Response: We have revised the aim of the paper.

  1. The abbreviations cannot be used in the Abstract without deciphering: line 24 - AChE and in line 375 - Full word ‘acetylcholinesterase’. The same with butyrlcholinesterase: line 24 – BchE and in line 369 - Full word ‘butyrlcholinesterase’.

Response: We have checked and explained all abbreviations.

  1. Lines 85-89 – The 'Results' section cannot start from the sentences like these ‘Phenolic compounds, found in plants, are the primary bioactive components with...'. It should start with the presentation of authors’ research results, and after it present their discussion and comparison with data from scientific sources.

Response: We have arranged the section based on the reviewer comments.

  1. Figure 3: There is no X-axis on the graph.

Response: We have already given X-axis (min: minute) on the graph.

  1. A moderatecheck is required for the English language and style through the text. For instance, line 195: ‘While scolignans, were only detected in bark’. The conclusion section should be more clearly formulated, for instance: In our study on Z. mauritiana, we found that homogenization-assisted extraction (HAE) with MeOH was more effective than maceration (MAE) with MeOH in terms of yielding higher phenolic content and antioxidant activity. We also discovered that MeOH and water extracts from Z. mauritiana leaves contain elevated levels of secondary metabolites such as spinosin and isospinosin … ‘

Response: We have checked the paper and also revised conclusion section.

  1. Lines 474-480:six different tests were used for studying antioxidant activity but their methodologies were not described.

Response: Appropriate references have been added to support the protocol used. For the sake of similarity, we have not included all details of the protocols as these have been previously published. However, all experimental details have been given in the supplemental materials

Reviewer 2 Report

Comments and Suggestions for Authors

The manuscript submitted for review under the title "Ziziphus mauritiana bark and leaves: Homogeneizer-assisted extraction phytochemical composition, in vitro bioassays in silico studies" represent experimental type of work. The selected topic is interesting for readers. The manuscript submitted for review describes the different extractions methods and solvents using on the quality and quantity chemical composition of Ziziphus mauritiana

Manuscript title is adequate to their contents.

 Some parts of the manuscript require additions according to the instructions and comments provided below:

 Abstract section:

Abstract give all information in short. Is good constructed. Please arrange keywords in alphabetical order.

 Some comments to manuscript as follows:

 Introduction section:

This part of the manuscript is very well written. I have no additional comments regarding this sub-section.

Material and method section:

Page 14, line 429: How many leaves and how many of bark pieces of Z. mauritiana were obtained and used for experimental purposes. Please add this information to manuscript text.

In sub-section “Material and methods” all procedure of the total polyphenols and total flavonoids analysis description is missing. Please add this description to manuscript text.

 Results and Discussion:

Figure 1 and 3. Please labeled identified phenolic compounds on the chromatogram picture. Please add as well retention time for all identified compounds.

 Figure 2 and 4 are missing. Figures numbering is wrong. Please verified it.

 All Tables are constructed well. There are clear and good presented.

 The whole discussion sub-section is written correctly. The obtained results were compared to those presented in the literature. I believe that this sub-section is presented properly.

 Conclusions:

 Can Authors add some practical conclusions of their obtained effect?

General opinion:  After carefully manuscript reading, I think, that presented experiment is a valuable. Manuscript should be publish after revision, according to my suggestions and points in Plants journal.

Author Response

Reviewer 2: Corrections are indicated in green color

The manuscript submitted for review under the title "Ziziphus mauritiana bark and leaves: Homogeneizer-assisted extraction phytochemical composition, in vitro bioassays in silico studies" represent experimental type of work. The selected topic is interesting for readers. The manuscript submitted for review describes the different extractions methods and solvents using on the quality and quantity chemical composition of Ziziphus mauritiana

Manuscript title is adequate to their contents.

 Some parts of the manuscript require additions according to the instructions and comments provided below:

 Abstract section:

Abstract give all information in short. Is good constructed. Please arrange keywords in alphabetical order.

Response: We have reordered the keywords in alphabetical order.

 Some comments to manuscript as follows:

 Introduction section:

This part of the manuscript is very well written. I have no additional comments regarding this sub-section.

Response: Thank you for your positive comments.

Material and method section:

Page 14, line 429: How many leaves and how many of bark pieces of Z. mauritiana were obtained and used for experimental purposes. Please add this information to manuscript text.

Response: We have added the infomations in the revised version.

In sub-section “Material and methods” all procedure of the total polyphenols and total flavonoids analysis description is missing. Please add this description to manuscript text.

Response: Total phenolic and flavonoid methods have been inserted in the revised text and  all experimental details have been given in the supplemental materials.

Results and Discussion:

Figure 1 and 3. Please labeled identified phenolic compounds on the chromatogram picture. Please add as well retention time for all identified compounds.

Response: We have checked them and labeled in the revised version.

 Figure 2 and 4 are missing. Figures numbering is wrong. Please verified it.

Response: We have renumbered all figures.

 All Tables are constructed well. There are clear and good presented.

 The whole discussion sub-section is written correctly. The obtained results were compared to those presented in the literature. I believe that this sub-section is presented properly.

Response: Thank you for your positive comments.

 Conclusions:

 Can Authors add some practical conclusions of their obtained effect?

Response: We have revised the conclusion section with practical conclusions

General opinion:  After carefully manuscript reading, I think, that presented experiment is a valuable. Manuscript should be publish after revision, according to my suggestions and points in Plants journal.

Response: Thank you for your positive comments.

Reviewer 3 Report

Comments and Suggestions for Authors

Dear authors, an insteresting paper. But may I suggest to enrich the materials and methods part and the introduction because it could be valuable for you to explain :

Your choices dealing with the extracting methods you choose. Why did you choose these methods, in particularly the HAE. What is the interests or the advantages of this methods according to you.

Please decribe it more precisely in the materials and methods part.

Why did you choose MeOH instead of EtOH. Because at the end in your conclusion you introduce the potential uses in functional foods, and according to me the use of this plant (Novel food in EU) and the Meoh could represent an obstacle for the use of this kind of extract in Functional foods.

Author Response

Reviewer 3: Corrections are indicated in turquoise color

Dear authors, an insteresting paper. But may I suggest to enrich the materials and methods part and the introduction because it could be valuable for you to explain :

Your choices dealing with the extracting methods you choose. Why did you choose these methods, in particularly the HAE. What is the interests or the advantages of this methods according to you.

Response: Extraction is a crucial step in the production of plant extracts. In this sense, in recent years, it has been reported that several green extraction techniques, such as microwave- or ultrasound-assisted extractions, are being replaced by traditional techniques such as maceration and Soxhlet. Among green extraction techniques, homogenizer-assisted extraction (HAE) is gaining increasing interest in phytochemical studies. HAE is a method in which the sample is rapidly rotated, introduced into the dispersing head in a straight line, and then pushed outward through slots in the rotor assembly. HAE has several advantages. For example, in this process the amount of solvent used is lower than in other techniques and thus ensures lower energy consumption and a shorter extraction time. This also provide by reducing the particle size of the plant material, which facilitates the release of phytochemicals into the medium.

Please decribe it more precisely in the materials and methods part.

Response: Appropriate references have been added to support the protocol used. For the sake of similarity, we have not included all details of the protocols as these have been previously published. However, all experimental details have been given in the supplemental materials

Why did you choose MeOH instead of EtOH. Because at the end in your conclusion you introduce the potential uses in functional foods, and according to me the use of this plant (Novel food in EU) and the Meoh could represent an obstacle for the use of this kind of extract in Functional foods.

Response: Methanol can extract a wider range of polar compounds compared to ethanol. The use of methanol to produce plant extracts in the field of drug development is based on its ability to efficiently extract a wide range of phytochemical compounds from plant material. Methanol is a commonly used solvent in laboratory environments due to its ability to dissolve both polar and non-polar compounds, making it suitable for the extraction of a wide range of plant components. In addition, it has a lower boiling point, higher volatility and higher extraction efficiency (depending on the desired secondary metabolite composition) compared to ethanol. During the extraction process, the methanol is typically evaporated or removed from the extract to yield the concentrated phytochemical compounds. The resulting extract can then be further processed, purified and examined for its potential medicinal properties in drug research and development. However, due to its toxic concerns, methanol has limited use in the nutraceutical field, and we have also revised our conclusion section at this point.

Round 2

Reviewer 1 Report

Comments and Suggestions for Authors

The authors significantly improved the manuscript. However, it is worth adding a comma in its name after the word "extraction"- Ziziphus mauritiana Lam. bark and leaves: extraction, phytochemical compositionin vitro bioassays and in silico studies’. 

A minor edit of the English language is necessary. Thus, in lines 201-203, it would be beneficial to combine these two sentences. It is also proposed to enhance the Conclusion section by rephrasing the first three sentences: "The comparison of extraction methods for Z. mauritiana bark and leaves showed that homogenization-assisted extraction (HAE) using MeOH was more effective, resulting in higher phenolic content and antioxidant activity compared to microwave-assisted extraction (MAE) with MeOH. It is important to note that the MeOH and water extracts from the leaves contained higher levels of secondary metabolites, such as spinosin and isospinosin. Based on the chemical structures of the most abundant compounds, we can infer that the flavonols present contributed to the antioxidant and enzyme inhibitory activities observed in the extracts..."

Comments on the Quality of English Language

A minor editing of the English language is required throughout the text.

Author Response

Reviewer 1: Corrections are indicated in yellow color

The authors significantly improved the manuscript. However, it is worth adding a comma in its name after the word "extraction"- ‘Ziziphus mauritiana Lam. bark and leaves: extraction, phytochemical composition, in vitro bioassays and in silico studies’. 

Response: We have corrected the title.

A minor edit of the English language is necessary. Thus, in lines 201-203, it would be beneficial to combine these two sentences. It is also proposed to enhance the Conclusion section by rephrasing the first three sentences: "The comparison of extraction methods for Z. mauritiana bark and leaves showed that homogenization-assisted extraction (HAE) using MeOH was more effective, resulting in higher phenolic content and antioxidant activity compared to microwave-assisted extraction (MAE) with MeOH. It is important to note that the MeOH and water extracts from the leaves contained higher levels of secondary metabolites, such as spinosin and isospinosin. Based on the chemical structures of the most abundant compounds, we can infer that the flavonols present contributed to the antioxidant and enzyme inhibitory activities observed in the extracts..."

Response: We have revised the sentences (201-203) and the three sentences have been revised in the conclusion section.